# Adaptive Multi-Modality Prompt Learning

**Zongqian Wu**
School of Computer Science and
Engineering, University of Electronic
Science and Technology of China
Chengdu, China
wkzongqianwu@gmail.com

**Yujing Liu**
Guangxi Key Lab of Multisource
Information Mining and Security,
Guangxi Normal University
Guilin, China
liuyujingcn@gmail.com

**Mengmeng Zhan**
School of Computer Science and
Engineering, University of Electronic
Science and Technology of China
Chengdu, China
memos.zhan@gmail.com

**Ping Hu**
School of Computer Science and
Engineering, University of Electronic
Science and Technology of China
Chengdu, China
chinahuping@gmail.com

**Xiaofeng Zhu**[*]
School of Computer Science and
Engineering, University of Electronic
Science and Technology of China
Chengdu, China
Shenzhen Institute for Advanced
Study, University of Electronic
Science and Technology of China
Shenzhen, China
seanzhuxf@gmail.com

## Abstract

Although current prompt learning methods have successfully been designed to effectively reuse the large pre-trained models without fine-tuning their large number of parameters, they still have limitations to be addressed, *i.e.*, without considering the adverse impact of meaningless patches in every image and without simultaneously considering in-sample generalization and out-of-sample generalization. In this paper, we propose an adaptive multi-modality prompt learning to address the above issues. To do this, we employ previous text prompt learning and propose a new image prompt learning. The image prompt learning achieves in-sample and out-of-sample generalization, by first masking meaningless patches and then padding them with the learnable parameters and the information from texts. Moreover, each of the prompts provides auxiliary information to each other, further strengthening these two kinds of generalization. Experimental results on real datasets demonstrate that our method outperforms SOTA methods, in terms of different downstream tasks. Our code is available at https://github.com/zongqianwu/PromptAMMPL.

## CCS Concepts

• **Vision and Language → Multimodal Fusion**.

---

[*]Corresponding author.

---

## Keywords

Vision-Language Models, Prompt Learning, Few-Shot Generalization, Multi-Modality Learning

**ACM Reference Format:**
Zongqian Wu, Yujing Liu, Mengmeng Zhan, Ping Hu, and Xiaofeng Zhu. 2024. Adaptive Multi-Modality Prompt Learning. In *Proceedings of the 32nd ACM International Conference on Multimedia (MM '24), October 28-November 1, 2024, Melbourne, VIC, Australia.* ACM, New York, NY, USA, 9 pages. https://doi.org/10.1145/3664647.3681485

## 1 Introduction

While large pre-trained vision-language models (such as CLIP [20]) have shown great potential for text-image alignment, prompt learning (PL) is popularly designed to learn diverse alignment for a large range of downstream tasks. Specifically, prompt learning techniques are designed to fine-tune the input data in order to better align images and texts within a shared space defined by a large pre-trained model. In particular, such a technique allows for reusing the pre-trained model without the need to tune its large number of parameters as well as fitting diverse downstream tasks [15, 16, 27].

Previous PL methods can be divided into three categories, *i.e.*, single-modality PL methods, non-interactive multi-modality PL methods, and interactive multi-modality PL methods. Specifically, single-modality PL methods design individual prompts to use the large pre-trained model. For instance, VPT [13] prompts the image for effectively using the pre-trained image encoder. Since single-modality PL methods only prompt one modality to ignore the prompt from the other modality, non-interactive multi-modality PL methods are designed to prompt both image modality and text modality. For instance, IVLP [14] learns two prompts for images and texts to show the generalization over known classes on unseen data [29, 30], in-sample generalization for short in this paper. However, previous non-interactive multi-modality PL methods are not able to effectively design prompts over widely unseen classes on unseen data, and thus easily resulting in the over-fitting issue. Recently,

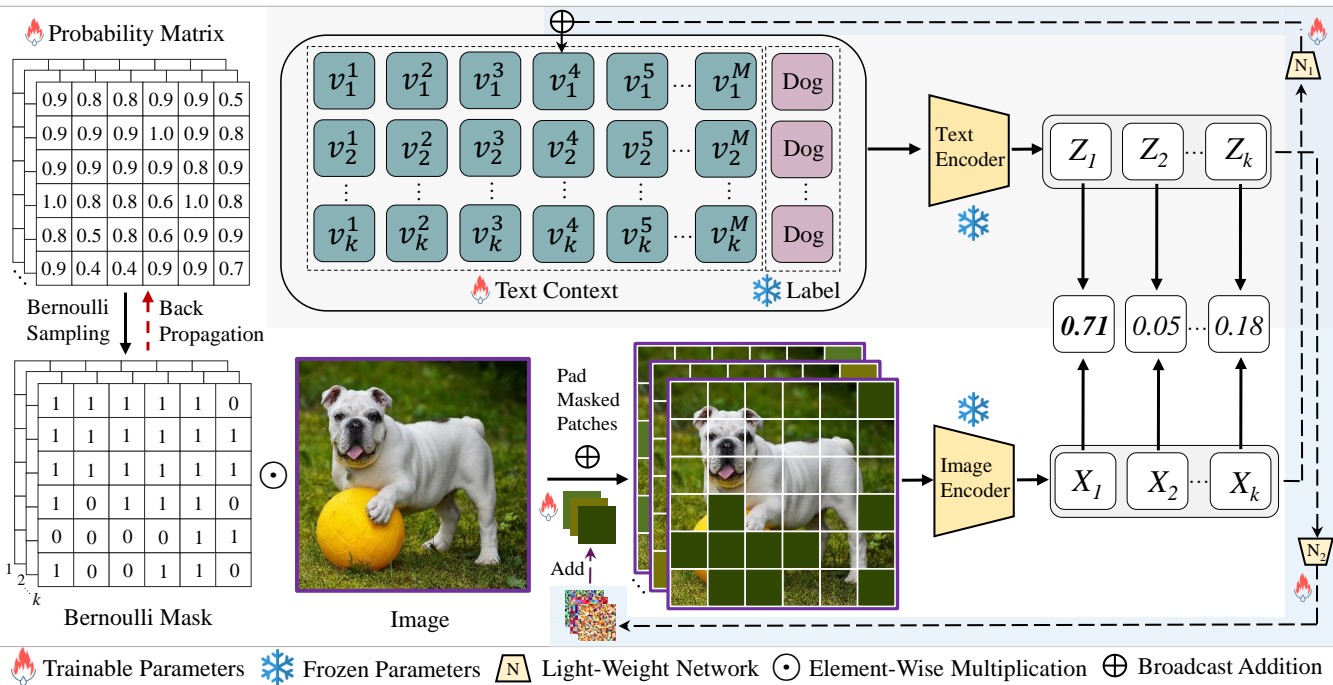

**Figure 1: The flowchart of the proposed AMMPL. The AMMPL consists of three modules, *i.e.*, Text Prompt Learning (light grey block), Image Prompt Learning (white block) and Adaptively Interactive Learning (light blue block). Specifically, text prompt learning outputs the text representation for text contexts by CoCoOp [29]. The proposed image prompt learning simultaneously detects and masks meaningless patches for every image, followed by generating the representation of this image, whose masked patches (*i.e.*, meaningless patches) are padded with learnable parameters and text information from adaptively interactive learning. In the proposed adaptively interactive learning, the output of image encoder (*i.e.*, image representation) is first fed into a light-weight network to output its new representation, which is then added to text context by the operation of broadcast addition. Meanwhile, the output of text encoder is first fed into a light-weight network to output its new representation, which is then added to the representation of masked patches.**

interactive multi-modality PL methods have been designed to learn two prompts as well as to obtain generalization over unseen classes on unseen data, out-of-sample generalization for short in this paper. For instance, MaPLe [14] achieves out-of-sample generalization by facilitating the interaction between two prompts. Although current multi-modality PL methods have widely been used on large pre-trained models, they still have limitations to be addressed.

Firstly, not all patches in the image are useful. Patches irrelevant to the image category (meaningless patches for short) for every image may result in adverse influence for determining the image category, ignored by previous PL methods. Recently, the literature (*e.g.*, VP [1]) adds noise to every patch so that it can reduce the influence of meaningless patches, but it may influence the meaningful patches for prompt learning. In real applications, a meaningless patch in one downstream task (or image category) may be useful for other downstream tasks (or image categories), so it is essential and challenging to handle meaningless patches.

Secondly, previous PL methods do not consider achieving both in-sample generalization and out-of-sample generalization. Specifically, downstream tasks include the classes or categories seen in the

training process as well as the classes unseen in the training process [22, 31]. Usually, previous PL methods robust to two kinds of generalization are well-known to be robust class shifts [29]. In the literature, single-modality and non-interactive multi-modality PL methods try to improve in-sample generalization only to easily result in the over-fitting issue. In contrast, interactive multi-modality PL methods focus on achieving out-of-sample generalization by ignoring in-sample generalization. Hence, previous prompt learning methods are not robust class shifts.

In this paper, we propose a new interactive multi-modality PL method, namely Adaptive Multi-Modality Prompt Learning (AMMPL) shown in Figure 1, to address the above issues. The proposed AMMPL consists of three modules, *i.e.*, text prompt learning, image prompt learning, and adaptive interactive learning. Specifically, we follow CoCoOp [29] to generate text representation for conducting text prompt learning. The proposed image prompt learning first learns a probability matrix and then employs Bernoulli sampling [3, 5] to detect and mask the meaningless patches for every category. The image with masked patches, which padded with learnable parameters and text information, are then fed into the image encoder. As a result, it addresses the first issue by reasonably

handling meaningless patches in the images. Moreover, the probability matrix sets meaningful patches with large probability and sets meaningless patches with small probability to improve the in-sample generalization ability. Bernoulli sampling makes the large values in the probability matrix possibly have small chance to be selected, such randomness improves the out-of-sample generalization ability. Hence, our image prompt learning addresses the second issue. Furthermore, our adaptively interactive learning conducts the information interaction between two modalities. Specifically, the light-weight network (*i.e.*, $N_1$ in Figure 1) propagates the image information to learn the text prompt, and thus promoting the effectiveness of text prompt learning. Similarly, the network $N_2$ in Figure 1 improves image prompt learning as well as explores two issues in previous methods. Hence, our adaptively interactive learning strengthens to solve two issues in previous PL methods.

Compared to previous methods, the main contributions of our method are two-fold. First, we propose a novel image prompt learning to solve two issues in previous PL methods. To our knowledge, it is the first work to explore the influence of meaningless patches for image prompt learning. Second, we investigate two light-weight networks (*i.e.*, two fully connected layers) for semantic and dimensional transformation of modality information, fostering efficient cross-modality interaction. As a result, each of them promotes the other. Moreover, the network from the text encoder strengthens the image prompt learning to solve the issues in previous PL methods.

## 2 Methodology

### 2.1 Motivations

The pre-trained vision-language model CLIP [20] includes two encoders, *i.e.*, the text encoder regarding Transformer as the backbone [23] for texts, and the image encoder employing ResNet [11] or Vision Transformer [6] as the backbone for images. Recently, prompt learning techniques have widely been used for pre-processing texts or images before feeding them into the encoder, aiming at improving the effectiveness of downstream tasks without tuning a large number of parameters in the large pre-trained models.

Inspired by the significant success of prompt learning in the field of natural language processing, the initial prompt learning techniques on CLIP aimed to adequately explore the potential of the text encoder through fine-tuning text modality, called text prompt learning. Since the focus of downstream tasks typically involves images, the latest PL methods focus on both text prompt learning and image prompt learning. However, previous PL methods still have two issues to be addressed. Firstly, they neglect the influence of meaningless patches in the images. As a result, CLIP may extract irrelevant representation for images to degrade the subsequent text-image alignment. Secondly, previous PL methods are not robust class shift, *i.e.*, not considering both in-sample generalization and out-of-sample generalization. As a result, they difficult to deal with diverse downstream tasks in real-world scenarios.

In this paper, we propose a new interactive multi-modality prompt learning method to address the above issues. Specifically, we first follow CoCoOp to perform text prompt learning in Section 2.2, and then design the image prompt learning in Section 2.3 and the adaptively interactive learning in Section 2.4 to address the above issues. We list the framework in Figure 1.

## 2.2 Text Prompt Learning

In CLIP, the text encoder takes fixed context tokens to make it inflexible for diverse downstream tasks, so text prompt learning techniques are designed to construct adaptable context tokens. For instance, CoCoOp [29] converts each context token of input text into a learnable vector to learn semantic context information that aligns with the specific downstream task. Hence, this paper follows CoCoOp to conduct text prompt learning.

Specifically, we first represent every context token by a learnable vector $v \in \mathbb{R}^l$ with the same length as the word embedding (*i.e.*, $l = 512$ in CLIP), and then replace the fixed context "a photo of a" by $M$ learnable vectors, *i.e.*, the learnable context $\{v_i^1, v_i^2, \ldots, v_i^M\}$. As a result, the text prompt of the $i$-th class is represented as $\mathbf{t}_i = \{v_i^1, v_i^2, \ldots, v_i^M, c_i\}$ where $c_i$ is the name of the $i$-th class, and the text prompt tensor for all classes is:

$$\mathbf{T} = \begin{bmatrix} v_1^1, & v_1^2, \cdots & v_1^M, & c_1 \\ v_2^1, & v_2^2, \cdots & v_2^M, & c_2 \\ \vdots & \vdots & \vdots & \vdots \\ v_k^1, & v_k^2, \cdots & v_k^M, & c_k \end{bmatrix}, \mathbf{T} \in \mathbb{R}^{k \times (M+1) \times l}, \quad (1)$$

where $k$ is the class number. We further input text prompt $\mathbf{T}$ into the text encoder TextEn$(\cdot)$ to obtain text representation, which is then fed into the text projection function TextProj$(\cdot)$ to obtain the final text representation $\mathbf{Z}$ by:

$$\mathbf{Z} = \text{TextProj}\left(\text{TextEn}\left(\mathbf{T}\right)\right), \quad \mathbf{Z} \in \mathbb{R}^{k \times d}, \quad (2)$$

where $d$ represents the dimension of the text representation. During the training process, the parameters of both TextEn$(\cdot)$ and TextProj$(\cdot)$ are frozen, while the parameters in $\mathbf{T}$ is adaptively adjusted to flexibly fix diverse tasks.

After conducting text prompt learning by Eq. (2), the text prompt $\mathbf{T}$ learns specific context for individual classes, thereby enabling the text encoder to extract fine-grained text representation for every class or category.

## 2.3 Image Prompt Learning

Besides the text encoder, it is crucial to consider image prompt learning because CLIP inherently includes both image input and text input. To address the two issues in previous PL methods, we should first partition the image into multiple patches, and then deal with meaningless patches before feeding the image including meaningless patches and meaningful patches into the image encoder. Meanwhile, it is also expected to obtain both in-sample generalization and out-of-sample generalization. To achieve this, the proposed image prompt learning consists of two steps, *i.e.*, patch mask and patch padding.

*2.3.1 Patch Mask.* As shown in Figure 3, an airplane is usually accompanied by the airport while the dog is accompanied by the streamlet. That is, the image category "airplane" is determined by the patches of the airplane and other patches relevant to the image category (*e.g.*, the patches relevant to the airport). In contrast, other patches provide little information to determine this category. Moreover, different image categories are determined by different patches. Hence, in the image prompt learning, we should 1) distinguish meaningless patches (*i.e.*, patches irrelevant to the image

category) from meaningful patches in the image, which determine the image class or category; and 2) design different prompts for different image categories.

Obviously, we may follow one of the backbones of CLIP (*i.e.*, Vision Transformer [6]) to first partition every image into $b \times b$ patches, and then detect meaningless patches based on the image partition. Motivated by MAE [10], we can first set a random matrix for every category to randomly mask all patches with binary values, and then conduct element-wise Hadamard product with the image to be the input of the image encoder. As a result, the random matrix (a.k.a., mask matrix) is adaptively updated to output the final results, *i.e.*, the meaningless patches with the binary value "0" for every category, and the image encoder outputs the representation of the masked patches distinguished from the representation of unmasked patches. However, the binary values in the random matrix make the back-propagation difficult.

In this paper, to address the above issue, we first generate a continuous probability matrix for every image category and then conduct the Bernoulli sampling on this probability matrix to obtain the mask matrix. By directly assigning the gradient obtained from the discrete mask matrix to the continuous probability matrix, our method makes the back-propagation available (Details in Section 2.5). As a result, after the optimization process, the mask matrix for every category can be obtained.

Specifically, after partitioning every image into $b \times b$ patches, we denote $\mathbf{P}_i \in \mathbb{R}^{b \times b}$ as the probability matrix of the $i$-th class. Every element in $\mathbf{P}_i$ is the probability of the patch belonging to a meaningless patch. Furthermore, the probability matrix for all $k$ categories/classes can be represented as a tensor, *i.e.*, $\mathbf{P} \in \mathbb{R}^{k \times b \times b}$. We conduct Bernoulli sampling on $\mathbf{P} \in \mathbb{R}^{k \times b \times b}$ to obtain the binary tensor $\mathbf{M} \in \mathbb{R}^{k \times b \times b}$ (Bernoulli mask for short) by:

$$m_i^{j,c} = \begin{cases} 1 & \text{with prob. } r = \text{Clamp}\left(p_i^{j,c}, 0, 1\right) \\ 0 & \text{with prob. } 1 - r, \end{cases} \quad (3)$$

where $r$ represents the probability of being sampled as 1, and Clamp$(\cdot, 0, 1)$ function restricts every element of $\mathbf{P}$ within the range between 0 and 1. The terms $j$ and $c$ respectively denote the row and column coordinates in the matrix. If the element in $\mathbf{M}$ is 0, the corresponding patch is masked.

We further perform the element-wise Hadamard product through broadcasting between $\mathbf{M} \in \mathbb{R}^{k \times b \times b}$ and the original image $\mathbf{I} \in \mathbb{R}^{b \times b \times u}$ to obtain:

$$\tilde{\mathbf{I}} = \mathbf{M} \odot \mathbf{I}, \qquad \tilde{\mathbf{I}} \in \mathbb{R}^{k \times b \times b \times u}, \quad (4)$$

where $u$ represents the channel number of images (*e.g.*, RGB channels). Based on the binary value in $\mathbf{M}$, the meaningful patches in $\tilde{\mathbf{I}}$ are preserved if the binary value is 1.

Eq. (3) solves the back-propagation issue caused in the random mask matrix by introducing a learnable probability tensor and Bernoulli sampling. As a result, Eq. (4) is able to distinguish meaningless patches from meaningful patches, which addresses the first issue in previous PL methods.

The optimal probability tensor $\mathbf{P}$ achieves the fine-grained sampling rate for the specific class, resulting in improving in-sample generalization ability. Moreover, due to the uncertainty introduced by the Bernoulli sampling, the obtained mask tensor $\mathbf{M}$ exhibits

diversity. Such randomness or uncertainty may improve the out-of-sample generalization ability [7]. Therefore, the patch mask addresses the second issue in previous PL methods by designing the dynamic probability tensor and the uncertain mask tensor to achieve robust class shifts.

Although the proposed patch mask addresses two issues present in previous PL methods, there is a significant gap in pixel values between the masked patches (*i.e.*, where pixel values are all 0) and other patches in the image. This significant difference in pixel values within the image easily results in more training iterations and requires more training data, which in turn makes difficult for the model to converge. Hence, we investigate to pad information into masked patches to address this issue.

*2.3.2* **Patch Padding**. Motivated by missing value padding techniques, an intuitive solution is to employ mean value padding to solve the pixel gap between masked and other patches in the image. However, the mean value for every masked patch makes the padding result in a lack of diversity, so it is difficult to learn different prompts for different image categories. In this paper, the patch padding step is designed to replace the mean value padding method for alleviating the pixel gap issue based on the goals: 1) task-relevant information adapted to image encoder; and 2) auxiliary information from text modality (Details in Section 2.4). To achieve the first goal, we investigate learnable parameters to pad masked patches in $\tilde{\mathbf{I}}$. This allows masked patches to contain task-relevant information from the image encoder.

Specifically, we propose to learn parameters for every category and the parameters of $i$-th class can be represented as $\mathbf{N}_i \in \mathbb{R}^{q \times q \times u}$, where $q \times q$ represents the size of the pixels in a patch. Therefore, the parameters of all classes can be represented as $\mathbf{N} \in \mathbb{R}^{k \times q \times q \times u}$. We then broadcast parameters separately to pad the masked patches of the corresponding classes by:

$$\hat{\mathbf{I}} = \tilde{\mathbf{I}}[\text{MASKED}] \oplus \mathbf{N}, \qquad \hat{\mathbf{I}} \in \mathbb{R}^{k \times b \times b \times u}, \quad (5)$$

where $\oplus$ is the broadcast addition. With the optimization of Eq. (5), supervision information from downstream tasks is embedded into masked patches, so that the masked patches are padded by the parameters and the pixel gap is alleviated. As a result, masked patches push the image encoder to detect different masked patches for specific image category.

We further input prompted image $\hat{\mathbf{I}}$ into the image encoder to obtain image representation $\mathbf{X}$ by:

$$\mathbf{X} = \text{ImageProj}\left(\text{ImageEn}\left(\hat{\mathbf{I}}\right)\right), \quad \mathbf{X} \in \mathbb{R}^{k \times d}, \quad (6)$$

where $d$ is the dimension of the text representation. Similar to text prompt learning, the parameters of the image encoder ImageEn$(\cdot)$ and the image projection function ImageProj$(\cdot)$ are frozen during the training process, while the parameters $\mathbf{P}$ and $\mathbf{N}$ are adaptively adjusted to flexibly fix diverse tasks or image input.

Since text prompt learning is independent on image prompt learning, their correlation is ignored. Actually, the correlation between two modalities has been demonstrated to provide auxiliary information to each other [14, 28, 29]. Hence, it is essential to consider their correlation for improving each of them.

## 2.4 Adaptively Interactive Learning

Previous interactive PL methods have studied the interactivity between two modalities by learning auxiliary information for other modality to improve generalization. However, they have the following issues to be addressed: 1) many previous methods are designed to obtain auxiliary information by handling complex internal structures within the model, and thus they need more training samples to achieve model convergence. For instance, MaPLe [14] and DPT [26] propagate auxiliary information across all layers to require more training samples. 2) Many previous methods transfer auxiliary information from one modality only by ignoring the auxiliary information from other modalities. For instance, CoCoOp [29] and DPT [26] propagate auxiliary information from the text modality to the image modality.

In this paper, the proposed adaptively interactive learning is designed to transfer auxiliary information from two modalities, to address the above issues. Specifically, given text representation $\mathbf{Z}$ and image representation $\mathbf{X}$, the representations of $i$-th class in $\mathbf{Z}$ and $\mathbf{X}$ are separately input into two light-weight networks to obtain the interaction information of $i$-th class as:

$$\begin{cases} \mathbf{E}_i = f_T^i(\mathbf{z}_i), & \mathbf{E}_i \in \mathbb{R}^{q \times q \times u} \\ h_i = f_I^i(\mathbf{x}_i), & h_i \in \mathbb{R}, \end{cases} \tag{7}$$

where $f_T^i(\cdot)$ and $f_I^i(\cdot)$ represent the $i$-th class text and image light-weight networks, respectively. These networks employ two fully connected layers for semantic and dimensional transformation of modality information, fostering efficient cross-modality interaction.

We then input the learned interaction information into two modalities. As a result, every context token of $i$-th class in text prompt $\mathbf{T}$ is updated to:

$$v_i^m = v_i^m + h_i, \tag{8}$$

where $m \in \{1, 2, ..., M\}$. Meanwhile, the learnable parameters of the $i$-th class in the image prompt is updated by:

$$\mathbf{N}_i = \mathbf{N}_i + \mathbf{E}_i. \tag{9}$$

Based on Eq. (8) and Eq. (9), our proposed adaptively interactive learning considers to provide auxiliary information to each modality from the other modality. Our proposed method only transfers auxiliary information into the input data, rather than all layers in many previous methods. As a result, the proposed method is able to make the model converge easily. Furthermore, Eq. (9) is used to pad the masked patches with the learnable parameters, benefiting the image encoder to use the relationship between text information and meaningful patches in the image. Since our proposed image prompt learning has been demonstrated to achieve both in-sample generalization and out-of-sample generalization in Section 2.3, the interaction between image information and text information (i.e., text information padded to masked patches) thus helps to strengthen these two kinds of generalization of our proposed multi-modality prompt learning.

Similar to CLIP, we further employ the text representation $\mathbf{Z}$ obtained from Eq. (2) and the image representation $\mathbf{X}$ obtained from Eq. (6) to compute the prediction probability by:

$$p(\hat{y} \mid \mathbf{X}) = \frac{\exp\left(\text{sim}\left(\mathbf{x}_{\hat{y}}, \mathbf{z}_{\hat{y}}\right) / \tau\right)}{\sum_{i=1}^k \exp\left(\text{sim}\left(\mathbf{x}_i, \mathbf{z}_i\right) / \tau\right)}, \tag{10}$$

where $\text{sim}(\cdot, \cdot)$ function represents cosine similarity score and $\tau$ is a temperature parameter. The prediction $\hat{y}$ corresponds to the class with the highest cosine similarity score. Moreover, as an image classification task, the standard cross-entropy loss $\mathcal{L}$ is the objective loss of our proposed method.

## 2.5 Optimization

In the optimization of image prompt learning, the Bernoulli mask $\mathbf{M}$ is non-continuous. This leads to $\frac{\partial \mathbf{M}}{\partial \mathbf{P}} = 0$ and $\frac{\partial \mathcal{L}}{\partial \mathbf{P}} = 0$. As a result, the standard back-propagation cannot be used to update the gradient of probability tensor $\mathbf{P}$. In this paper, we propose a simple yet effective method to learn gradients of the probability tensor $\mathbf{P}$.

Specifically, during the forward propagation, we reconstruct the computational graph of the Bernoulli mask $\mathbf{M}$ as:

$$\mathbf{M} = \text{Detach}(\mathbf{M} - \mathbf{P}) + \mathbf{P}, \tag{11}$$

where $\text{Detach}(\cdot)$ detaches tensors from the computational graph. During the back-propagation, we directly propagate the gradient of the Bernoulli mask $\mathbf{M}$ to the tensor $\mathbf{P}$ as:

$$\frac{\partial \mathcal{L}}{\partial \mathbf{P}} = \frac{\partial \mathcal{L}}{\partial \mathbf{M}}. \tag{12}$$

Eq. (11) reconstructs the Bernoulli mask by two components, i.e., the probability tensor $\mathbf{P}$ and the difference $\text{Detach}(\mathbf{M} - \mathbf{P})$ between the probability tensor and the Bernoulli mask, where $\mathbf{P}$ is differentiable. Therefore, by forwarding the reconstructed Bernoulli mask to compute the loss, the gradient obtained from the loss calculation can be back-propagated to $\mathbf{P}$ through Eq. (12).

Based on Eq. (12), our proposed method can efficiently conduct gradient update on the continuous probability tensor $\mathbf{P}$ by indirectly updating the discrete Bernoulli mask $\mathbf{M}$ during the back-propagation process.

## 3 Experiments

## 3.1 Experimental Settings

We evaluate our AMMPL with 7 comparison methods in terms of one in-sample task (i.e., few-shot learning) and two out-of-sample tasks (i.e., base-to-novel generalization and cross-data evaluation) on 9 benchmark datasets. The used datasets include four fine-grained datasets (i.e., OxfordPets [19], Flowers102 [18], Food101 [2], and FGVCAircraft [17]), one generic-objects dataset, i.e., Caltech101 [8], one satellite-image dataset, i.e., EuroSAT [12], one texture dataset, i.e., DTD [4], one action recognition dataset, i.e., UCF101 [21], and one scene recognition dataset, i.e., Sun397 [25]. The comparison methods include three single-modality PL methods (i.e., DLP [14], VPT [13], and VP [1]), one non-interactive multi-modality PL method, i.e., IVLP [14], and three interactive multi-modality PL methods, i.e., CoCoOp [29], DPT [26], and MaPLe [14].

We follow CoCoOp [29] to implement our method as well as apply prompt tuning to the pre-trained ViT-B/16 in CLIP [20]. In our experiments, we follow the literature [31] to extend our proposed

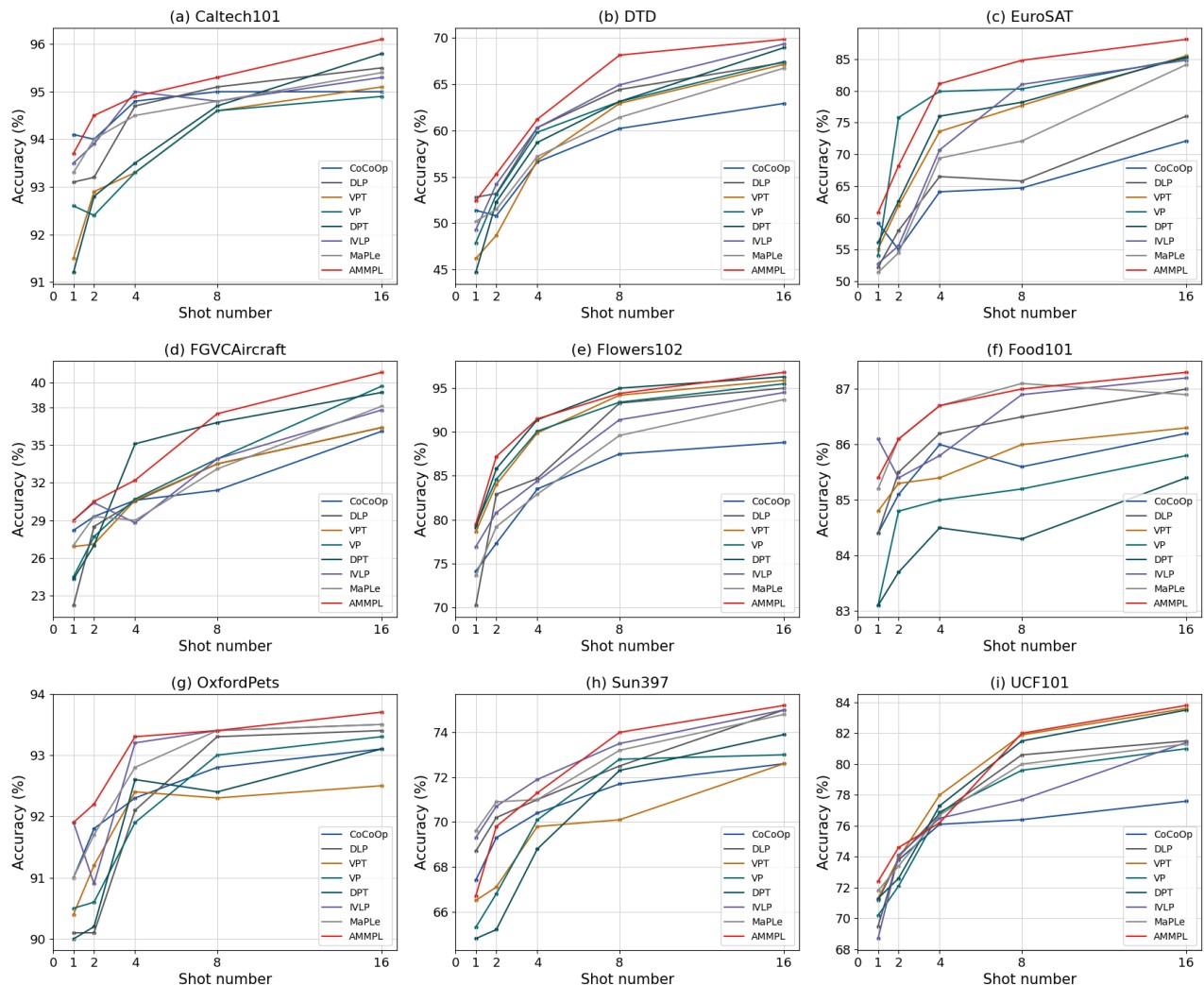

**Figure 2: Classification accuracy over 3 runs of all methods at different shot numbers, *i.e.*, {1, 2, 4, 8, 16} on all datasets.**

model into two versions, *i.e.*, class-specific AMMPL for evaluating in-sample generalization ability in Section 3.2 and alternative AMMPL for evaluating out-of-sample generalization ability in Section 3.3. In particular, all categories in alternative AMMPL employ the same context in text prompt learning, the common probability matrix in image prompt learning, and the same light-weight networks in adaptively interactive learning.

All experiments of all methods are conducted on as server with NVIDIA Tesla V100S (32GB memory each). In our method, the initialized probability matrix $\mathbf{P}$ is sampled from a Gaussian distribution, with the mean value set within the range of {0.80, 0.90, 0.93, 0.95, 0.97} and a standard deviation of 0.05. The initialization of $\mathbf{T}$ is the vector representation of "a photo of a", and noise $\mathbf{N}$ is initialized from a Gaussian distribution (*i.e.*, mean = 0, standard deviation = 0.02). Furthermore, to ensure fair comparison, we set the parameters of all comparison methods in accordance with the original literature to achieve their best performance outputs.

## 3.2 In-sample Few-shot Generalization

We evaluate the in-sample generalization of all methods by reporting the results of few-shot learning with different shot numbers (*i.e.*, 1, 2, 4, 8, and 16) in Figure 2.

The proposed AMMPL achieves the best performance. First, our method outperforms all single-modality PL methods (*i.e.*, DLP, VPT, and VP) and the non-interactive multi-modality PL method (*i.e.*, IVLP). For example, the proposed AMMPL averagely improves by 1.62%, 2.30%, 2.45%, 1.73%, and 1.66% respectively, compared with the best method IVLP, in terms of 1-shot, 2-shot, 4-shot, 8-shot, and 16-shot on all datasets. This contributes to that the probability tensor $\mathbf{P}$ achieves the optimal sampling rate. Hence, the Bernoulli mask $\mathbf{M}$ can mask out meaningless patches and our proposed patch padding can pad useful information. Both of them guarantee the image encoder in our method to improve the in-sample generalization ability. Second, our method outperforms interactive multi-modality

**Table 1: Classification accuracy (mean and standard deviation) over 3 runs of all interactive multi-modality methods (with ViT-B/16) in terms of generalization from base-to-novel classes at 16-shot learning on all datasets. Note that, "Base", "Novel", and "HM", respectively, indicate the classification accuracy of the base classes, the novel classes, and the harmonic mean.**

| (a) Caltech101 | Base | Novel | HM |
|---|---|---|---|
| CoCoOp | 97.80(0.1) | 93.00(0.1) | 95.34(0.1) |
| MaPLe | 97.89(1.4) | 94.30(0.4) | 96.06(0.7) |
| AMMPL | **97.99(0.1)** | **94.59(0.1)** | **96.25(0.1)** |

| (b) DTD | Base | Novel | HM |
|---|---|---|---|
| CoCoOp | 77.30(0.5) | 54.57(1.2) | 63.97(0.7) |
| MaPLe | **79.37(2.1)** | 53.80(7.5) | 64.13(3.2) |
| AMMPL | 78.33(1.5) | **58.43(3.1)** | **66.93(2.0)** |

| (c) EuroSAT | Base | Novel | HM |
|---|---|---|---|
| CoCoOp | 85.63(2.7) | 60.33(4.5) | 70.79(3.4) |
| MaPLe | 93.60(1.1) | 65.47(8.2) | 77.05(2.0) |
| AMMPL | **94.10(2.0)** | **67.39(6.3)** | **78.54(3.0)** |

| (d) FGVCAircraft | Base | Novel | HM |
|---|---|---|---|
| CoCoOp | 34.37(0.5) | 32.70(1.0) | 33.51(0.6) |
| MaPLe | 35.46(1.9) | 34.61(4.5) | 35.03(2.6) |
| AMMPL | **35.69(1.6)** | **35.91(1.3)** | **35.80(1.4)** |

| (e) Flowers102 | Base | Novel | HM |
|---|---|---|---|
| CoCoOp | 94.97(1.2) | 71.43(1.4) | 81.53(1.3) |
| MaPLe | **95.47(0.2)** | 73.33(2.3) | 82.94(0.4) |
| AMMPL | 94.90(1.1) | **74.61(1.3)** | **83.54(1.2)** |

| (f) Food101 | Base | Novel | HM |
|---|---|---|---|
| CoCoOp | 90.67(0.2) | 91.27(0.6) | 90.96(0.3) |
| MaPLe | 90.72(0.1) | 92.07(0.1) | 91.39(0.1) |
| AMMPL | **90.90(0.1)** | **92.10(0.2)** | **91.50(0.1)** |

| (g) OxfordPets | Base | Novel | HM |
|---|---|---|---|
| CoCoOp | 95.20(0.4) | 97.89(0.1) | 96.52(0.2) |
| MaPLe | 95.60(0.3) | 97.63(0.3) | 96.60(0.3) |
| AMMPL | **96.11(0.3)** | **98.03(0.1)** | **97.31(0.1)** |

| (h) Sun397 | Base | Novel | HM |
|---|---|---|---|
| CoCoOp | **81.27(0.5)** | **78.90(0.7)** | **80.07(0.6)** |
| MaPLe | 80.50(0.2) | 78.10(0.2) | 79.28(0.2) |
| AMMPL | 81.02(0.3) | 78.49(0.3) | 79.73(0.3) |

| (i) UCF101 | Base | Novel | HM |
|---|---|---|---|
| CoCoOp | 81.27(0.5) | 73.77(2.5) | 77.34(0.9) |
| MaPLe | **83.87(0.3)** | 76.20(1.8) | **79.85(0.5)** |
| AMMPL | 82.58(0.5) | **76.72(1.2)** | 79.54(0.7) |

**Table 2: Classification accuracy (mean and standard deviation) over 3 runs of all interactive multi-modality methods (with ViT-B/16) in terms of cross-data evaluation with different shot numbers (*i.e.*, 1-shot, 8-shot, and 16-shot) on all datasets.**

| Method | Shot | Source (Food101) | Target | | | | | | | |
|---|---|---|---|---|---|---|---|---|---|---|
| | | | Caltech101 | DTD | EuroSAT | FGVCAircraft | Flowers101 | OxfordPets | Sun397 | UCF101 |
| CoCoOp | 1 | 84.90(0.5) | 84.70(4.0) | 31.83(1.5) | 38.27(5.0) | 11.40(4.5) | 55.37(8.8) | 74.80(4.2) | 51.27(2.2) | 58.80(0.9) |
| MaPLe | 1 | 82.97(3.9) | 85.50(7.6) | 30.37(3.6) | 45.50(2.6) | 10.47(4.4) | 55.34(5.5) | 75.13(6.9) | 50.20(4.7) | 55.95(3.9) |
| AMMPL | 1 | **85.17(0.7)** | **87.23(1.3)** | **35.30(1.7)** | **45.90(1.4)** | **15.53(3.0)** | **59.70(3.0)** | **79.03(4.0)** | **54.53(0.4)** | **60.55(3.6)** |
| CoCoOp | 8 | 86.90(0.4) | 89.77(2.2) | 28.83(0.8) | 43.17(3.0) | 16.97(2.3) | 59.40(2.4) | 74.63(5.5) | 54.67(1.9) | 62.88(1.4) |
| MaPLe | 8 | 86.73(0.4) | 89.60(1.4) | 37.57(6.2) | 45.90(5.1) | 16.40(9.4) | 65.70(4.9) | 78.87(6.3) | 54.63(2.6) | **62.97(2.4)** |
| AMMPL | 8 | **87.07(0.2)** | **89.90(1.5)** | **39.73(2.8)** | **46.03(1.9)** | **19.50(2.4)** | **66.01(2.6)** | **79.00(3.4)** | **55.48(2.1)** | 61.85(3.9) |
| CoCoOp | 16 | 87.03(0.2) | 90.00(1.7) | 41.50(2.6) | 45.57(3.9) | 18.53(1.8) | 65.70(4.93) | 80.77(1.1) | 59.50(1.4) | 62.40(0.9) |
| MaPLe | 16 | 87.20(0.3) | 90.67(1.0) | 41.33(2.7) | **47.60(2.7)** | 18.53(3.8) | 66.10(1.92) | 80.70(8.2) | **61.33(3.2)** | **64.23(2.2)** |
| AMMPL | 16 | **87.30(0.3)** | **92.48(2.1)** | **42.17(0.8)** | 47.04(5.3) | **20.01(2.1)** | **66.47(1.27)** | **81.30(2.9)** | 60.80(1.2) | 62.77(2.8) |

**Table 3: Classification accuracy (mean and standard deviation) of AMMPL with different components (C1, C2 and C3 represent the patch mask, patch padding, and adaptively interactive learning components, respectively) at 1-shot on all datasets and the bold number represents the best results in the whole column.**

| Combo | Caltech101 | DTD | EuroSAT | FGVCAircraft | Flowers102 | Food101 | OxfordPets | Sun397 | UCF101 |
|---|---|---|---|---|---|---|---|---|---|
| C1 | 92.33(0.6) | 49.47(1.9) | 51.47(3.1) | 26.90(0.2) | 73.57(1.0) | 83.13(0.8) | 90.97(0.7) | 67.20(0.5) | 69.30(0.9) |
| C3 | **94.13(0.4)** | 51.41(1.4) | 59.22(2.9) | 28.25(0.4) | 74.14(0.7) | 84.40(0.6) | 91.67(0.5) | 67.43(0.7) | 71.23(0.6) |
| C1+C2 | 93.20(0.2) | 49.20(0.3) | 57.47(7.1) | 28.30(1.2) | 75.53(2.9) | 84.23(1.3) | 91.30(0.4) | **68.27(0.4)** | 70.13(2.5) |
| C1+C2+C3 | 93.71(0.3) | **52.42(1.3)** | **60.85(2.4)** | **29.12(0.1)** | **79.79(0.8)** | **85.45(0.6)** | **91.93(0.4)** | 66.89(0.2) | **72.47(0.7)** |

PL methods (*i.e.*, CoCoOp, DPT, and MaPLe) by a large margin since our method provides auxiliary information for individual modalities. The reason is that they place excessive emphasis on the interaction between modalities, resulting in sub-optimal model fitting to training samples.

## 3.3 Out-of-sample Generalization

*3.3.1 **Generalization from Base-to-Novel Classes**.* We investigate out-of-sample generalization by first training all methods (*i.e.*, our proposed AMMPL, CoCoOp and MaPLe) on the base classes

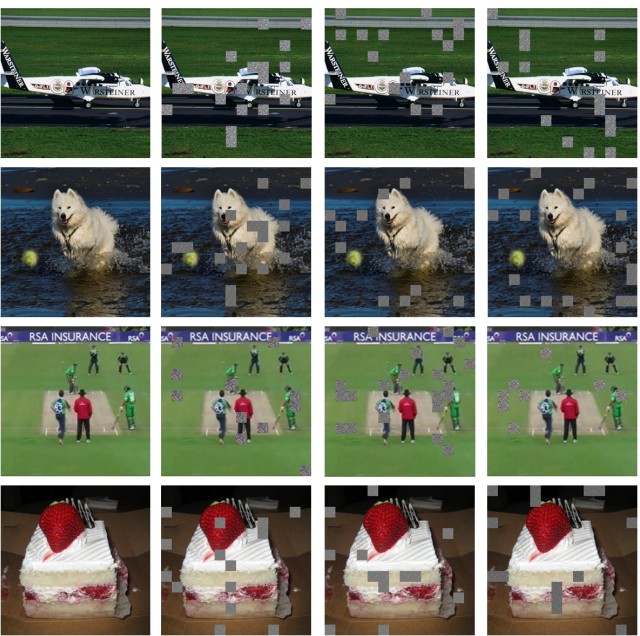

**Figure 3: Visualization of masked patches by our method with increased iteration. Note that, the first column is the training images, and the iterations increase from the 2nd column to the 4th column.**

and then evaluating them on the novel classes. Note that, both the base classes and the novel classes come from the same dataset.

Based on the second column of Table 1, our method shows average improvement of 1.20% over the best method MaPLe due to the reasons as follows: 1) the probability tensor **P** generates diverse presentation for masked patches due to its randomness, which serves as a regularization term to improve the out-of-sample generalization [7]; and 2) our method conducts interactions on the input side of prompts effectively alleviates over-fitting issues. Similar to the result of few-shot learning, the first column of Table 1 verifies our method to achieve in-sample generalization again. Furthermore, we follow the literature [24] to evaluate the harmonic mean [9] between in-sample generalization and out-of-sample generalization. Based on the result in the third column of Table 1, our proposed method simultaneously achieves two kinds of generalization.

*3.3.2 Cross-data Evaluation.* We follow [29] to conduct cross-data evaluation to further evaluate the out-of-sample generalization of the proposed AMMPL. This requires model training on one dataset and model evaluation on other datasets. Specifically, we first train all interactive prompt learning methods (*i.e.*, our AMMPL, CoCoOp and MaPLe) on the source dataset (*i.e.*, Food101) with different shot numbers (*i.e.*, 1, 8, and 16) and then test these methods on the remaining 8 datasets. We report the results in Table 2.

As a result, the proposed AMMPL averagely improves by 3.55%, 0.69%, and 0.28% respectively, compared with the best method MaPLe, in terms of 1-shot, 8-shot and 16-shot on all datasets. Obviously, our proposed AMMPL shows significant advantages when

the shot numbers are small. However, with the increase of the shot numbers, its performance is gradually approached by MaPLe. The reason is that our AMMPL only interacts within the input data to obtain the trade-off between in-sample generalization and out-of-sample generalization. In contrast, MaPLe interacts in all coding layers to gradually improve out-of-sample generalization with the increase of the shot numbers. However, MaPLe achieves out-of-sample generalization only.

## 3.4 Ablation Studies

The key components of the proposed AMMPL include patch mask, patch padding, and adaptively interactive learning. To demonstrate the effectiveness of individual components, we investigate the performance of in-sample generalization task (*i.e.*, few-shot learning) using different combinations of these components on all datasets.

We present few-shot learning results in Table 3. First, our method with all components improves on average by 2.19%, compared with the methods with one component only. This indicates that both image prompt learning and adaptively interactive learning are essential in our method. Second, image prompt learning outperforms adaptively interactive learning because the latter is used to strengthen the two kinds of generalization by providing auxiliary information. Third, the patch mask shows weak performance but it reports good performance while combing with patch padding. This indicates that the pixel gap has an adverse impact on the model learning. After the patch padding, the pixel gap is alleviated so that the model learning is robust to specific tasks.

Additionally, we visualize masked patches to demonstrate the effectiveness of the proposed patch mask in Figure 3. As a result, meaningless patches in the image are gradually masked with the increase of training iterations. For instance, the mask is gradually shifted from the dog (*i.e.*, meaningful patches) to its background (*i.e.*, meaningless patches) in the second row of Figure 3.

## 4 Conclusion

In this paper, we proposed an adaptive multi-modality prompt learning consisting of text prompt learning, image prompt learning, and adaptively interactive learning. To do this, we followed CoCoOp to perform text prompt learning. We also proposed image prompt learning to handle meaningless patches in the image as well as to achieve in-sample generalization and out-of-sample generalization. We further proposed adaptively interactive learning to strengthen these two kinds of generalization by achieving interactivity between texts and images. Extensive experimental results on real datasets showed that our method achieves supreme performance, compared to previous SOTA prompt learning methods.

## 5 Acknowledgments

This work was supported in part by National Key Research and Development Program of China under Grant 2022YFA10041000, Natural Science Foundation of Guangdong Province of China under Grant 2024A1515011381, and Guangxi "Bagui" Teams for Innovation and Research, China.

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
