# OpenReview forum: "Adaptive Multi-Modality Prompt Learning"
_acmmm.org/ACMMM/2024/Conference — MM2024 Poster_

### Official Review · Reviewer_dvBP · 2024-05-06

**Rating:** 4
**Confidence:** 3

**Summary:**

In this paper, an adaptive multimodal prompt learning method is proposed to address the limitations of current prompt learning methods. It introduces text prompt learning and image prompt learning by masking meaningless patches in order to achieve both in-sample and out-of-sample generalization. Experimental results show that the proposed method outperforms state-of-the-art methods in a variety of downstream tasks.

**Strengths:**

1. AMMPL innovatively proposes masking meaningless regions to reduce their negative impact on prompt learning.
2. AMMPL employs a lightweight network to combine visual and textual information, thereby improving generalization.

**Limitations:**

1. According to Table 3, when comparing the results of just having patch masking and just having interactive learning components, it appears that patch masking might have side effects. Moreover, according to Figure 3, masking during iterations also masks out meaningful parts. So, does this type of masking truly have an effective impact?
2. AMMPL first obtains a probability matrix through CLIP, then proceeds to mask based on this matrix. It heavily relies on CLIP's own image recognition capabilities. If the accuracy of CLIP's image recognition is inherently low—for example, using a ResNet50 image encoder where CLIP achieves only a 23% accuracy on the Aircraft dataset—would this significantly impair the effectiveness of the masking process?
3. AMMPL has not been compared with PromptSRC "Self-regulating Prompts: Foundational Model Adaptation without Forgetting" [ICCV 2023]. In that paper, the base-to-new results, such as a 98.07% accuracy for Flower102, exceed AMMPL's 94.9%, and an 83.37% accuracy for DTD surpasses AMMPL's 78.33%.

**Suitability:**

3

---

### Official Review · Reviewer_xBjN · 2024-05-20

**Rating:** 2
**Confidence:** 3

**Summary:**

This paper introduces an adaptive multi-modality prompt learning approach to address limitations in current prompt learning methods. The proposed method aims to effectively reuse large pre-trained models without fine-tuning a large number of parameters while considering the impact of meaningless patches in images and balancing in-sample and out-of-sample generalization. By combining text prompt learning with a new image prompt learning technique, the approach masks meaningless patches in images and replaces them with learnable parameters and information from texts to improve generalization. Additionally, the prompts provide auxiliary information to each other, enhancing both in-sample and out-of-sample generalization. Experimental results on real datasets demonstrate that the proposed method outperforms state-of-the-art methods across different downstream tasks.

**Strengths:**

1. The paper is well-written and easy to follow.

2. The idea is novel. The approach demonstrates superior performance across various downstream tasks compared to baseline methods.

**Limitations:**

- MaPLe, a work from 2022, is referenced in the paper, but there seems to be a lack of comparison with the most recent studies in the field.
 It would be beneficial to include a comparative analysis with the latest work to better situate the contributions of this paper.

- The concept of patch redundancy is intriguing; however, the rationale behind how it promotes both in-sample generalization and out-of-sample generalization is not clearly articulated. The authors appear to draw inspiration from the VP (Variable Projection) approach, but the distinctions between their method and VP are not well explained.

- The proposed method requires multiple forward passes for image processing. It is suggested that the authors provide a comparative analysis in terms of memory consumption and training speed, which are critical factors for evaluating the practicality of the approach.

- For both in-sample generalization and out-of-sample generalization, it is recommended that the paper presents comprehensive metrics to facilitate comparison between different methods. The results from various datasets indicate that the effectiveness of the proposed method is not consistent across them, with improvements being rather limited.

**Suitability:**

3

---

### Official Review · Reviewer_aKAT · 2024-05-24

**Rating:** 5
**Confidence:** 2

**Summary:**

This paper proposes a new interactive multi-modality PL method, namely AMMPL, considering the adverse impact of meaningless patches in every image. It addresses the problem of in-sample generalization and out-of-sample generalization based on Bernoulli sampling. Extensive experiments on nine datasets prove its effectiveness.

**Strengths:**

1. The paper might be the first effort (the authors claimed) to explore the influence of meaningless patches for image prompt learning. The idea is novel and interesting.
2. The designed light-weight networks promote  the other to foster efficient cross-modality interaction.
3. Extensive experiments and analyses demonstrates the superiority of AMMPL.
4. The source code is available.

**Limitations:**

1. The experimental evaluation of the paper mainly focuses on image classification accuracy, and the proposed multi-modality prompt lacks an assessment of the quality of multi-modality interaction.
2. The paper lacks an analysis of failure cases. Although the experimental results demonstrate the superiority of the method, it fails to provide a detailed analysis of the circumstances under which the method performs poorly and the possible reasons for these shortcomings.
3. The paper mentions generating a continuous probability matrix for every image category. Further elaboration is needed on how this is applied to a single image to simultaneously detect and mask meaningless patches.

**Suitability:**

3

---

### Meta-Review · Area_Chair_1C2x · 2024-06-28

**Recommendation:** Accept (Poster)
**Confidence:** 3

**Metareview:**

This paper study the prompt learning in the literature of large pre-trained models, claiming that the current techniques did not consider rhe adverse impact of meaningless patches and its generalisation performance. The authors suggest to combine text prompt learning with a new image prompt learning technique, which masks meaningless patches in images and replaces them with learnable parameters and information from texts to improve generalization.

The combination of image prompt and text prompt seems interesting, and all the reviewers also think the paper has some novelty. The reviewers raise questions about the new baselines, comparison with previous works, as well as more analysis about the mechanism. The authors have tried to address these concerns during rebuttal. All the reviewers tend to accept the paper and I also tend to accept the paper.

---

### Meta-Review · Senior_Area_Chairs · 2024-07-10

**Recommendation:** Accept (Poster)
**Confidence:** 4

**Metareview:**

This paper received mixed ratings initially. After rebuttal, all the reviewers tend to accept the paper. SAC and AC agree with reviewers and recommend acceptance of the paper.